# Machine and Deep Learning in the Evaluation of Selected Qualitative Characteristics of Sweet Potatoes Obtained under Different Convective Drying Conditions

Krzysztof Przybył [1,*] , Franciszek Adamski [1], Jolanta Wawrzyniak [1] , Marzena Gawrysiak-Witulska [1], Jerzy Stangierski [2] and Dominik Kmiecik [3,*]

1 Department of Dairy and Process Engineering, Faculty of Food Science and Nutrition, Poznań University of Life Sciences, Ul. Wojska Polskiego 31, 60-624 Poznań, Poland

2 Department of Food Quality and Safety Management, Faculty of Food Science and Nutrition, Poznań University of Life Sciences, Ul. Wojska Polskiego 31, 60-624 Poznań, Poland

3 Department of Food Technology od Plant Origin, Faculty of Food Science and Nutrition, Poznań University of Life Sciences, Ul. Wojska Polskiego 31, 60-624 Poznań, Poland

* Correspondence: krzysztof.przybyl@up.poznan.pl (K.P.); dominik.kmiecik@up.poznan.pl (D.K.); Tel.: +48-61-848-79-32 (K.P.)

**Abstract:** This paper discusses the use of various methods to distinguish between slices of sweet potato dried in different conditions. The drying conditions varied in terms of temperature, the values were: 60 °C, 70 °C, 80 °C, and 90 °C. Examination methods included instrumental texture analysis using a texturometer and digital texture analysis based on macroscopic images. Classification of acquired data involved the use of machine learning techniques using various types of artificial neural networks, such as convolutional neural networks (CNNs) and multi-layer perceptron (MLP). As a result, in the convective drying, changes in color darkening were found in products with the following temperature values: 60 °C (L = 83.41), 70 °C (L = 81.11), 80 °C (L = 79.02), and 90 °C (L = 75.53). The best-generated model achieved an overall classification efficiency of 77%. Sweet potato dried at 90 °C proved to be completely distinguishable from other classes, among which classification efficiency varied between 61–83% depending on the class. This means that image analysis using deep convolutional artificial neural networks is a valuable tool in the context of assessing the quality of convective-dried sweet potato slices.

**Keywords:** artificial neural networks; convolutional neural networks; machine learning; deep learning; sweet potato; convective drying

## 1. Introduction

As a rich source of carbohydrates, sweet potatoes (*Ipomoea batatas*) are a basic component of the diet of a significant number of people living in warm climatic zones [1]. Their consumption may positively affect human health [2]. The relatively low glycemic index of their carbohydrates makes sweet potatoes often recommended in the diet of diabetics. Sweet potatoes also contain many minerals (Ca, P, K, Na, Mg, S, Cl, Fe, J and smaller amounts of Mn, Cu, Mo, Se), as well as vitamins B, C, and E [3–5]. Violet varieties of sweet potatoes are also rich in pro-health anthocyanins, which are strong antioxidants [6] that indirectly inhibit the secretion of glucose, decreasing its blood levels [7]. Cultivars with orange-colored flesh containing significant amounts of carotenoids are an important source of provitamin A [8], while those with yellow flesh are characterized by the presence of lutein. On the other hand, red sweet potatoes are a rich source of lycopene, which may reduce the risk of heart disease and some cancers. Due to its chemical composition and properties, sweet potatoes are considered one of the healthiest foods in the world [1] and are recommended by NASA as a superfood for space expeditions [9].

However, sweet potatoes are very unstable and difficult to transport and store over long periods of time due to their high moisture content and post-harvest metabolic activity and thin, permeable and brittle outer skin. For this reason, the transport of fresh sweet potatoes to places far away from their cultivation areas is significantly hampered by the loss of water from their tubers. Pre-treatment of sweet potatoes aimed at preserving this raw material minimizes the risk of their spoilage or quality deterioration. One of the most frequently used methods of maintaining the health-promoting properties of sweet potatoes is convective drying [10,11]. This technique is a development of a traditional method of food dehydration, namely sun drying. The dynamics of the process is conditioned by two mechanisms of water evaporation. The distinction is due to the formation of a crust on the surface of the dried material, which prevents the penetration of water vapor from the dried material into the drying air and heat in the opposite direction. In this context, it is important to distinguish between the evaporation from the surface of the material and the evaporation taking place inside it [12]. It allows the use of repeatable parameters in the drying process and thus the ability to obtain dried material with repeatable quality characteristics with low equipment and technical requirements [13]. Food waste has recently been reported to be a serious economic problem. It is estimated that in the USA alone, the amount of food lost is close to 103 million tons, of which 39.8 million tons are wasted by the food processing industry [14]. For this reason, efforts are made by the food production industry to implement solutions that will help alleviate hunger, a phenomenon that is still present in many underdeveloped countries. Such efforts involve the use of modern technological solutions to eliminate losses leading to the inefficient use of processed raw materials. Choosing non-invasive technologies with the lowest possible costs is particularly important, as well as the quality of obtained products. Growing consumer awareness puts pressure on food producers to supply products that meet certain standards (norms) [15], forcing them to search for methods allowing for effective control and assessment of the quality of manufactured food products. In connection with continuous development and actions taken to modernize and automate food production processes, methods used to assess the quality of raw materials and food products have also been evolving, and now also encompass analysis of sensory characteristics, including color [16–18], shape, and texture, using instrumental methods [19,20]. The adequate analysis of sensory characteristics of food products based on digital photos may be a key factor in the development of these methods. In this context, artificial intelligence methods appear to be the right tool to support various decision-making and food processing and preservation processes [21].

Evaluation of material based on digital photos requires appropriate statistical processing, necessitating the use of computational systems and complex statistical tools for data analysis. Of particular note is the use of artificial neural networks, including deep artificial neural networks, which enable the classification of photos based on objects visible in them. Neural networks as a subset of machine learning are increasingly used in supporting and solving many decision problems [22–24] and in additive manufacturing [25–27]. They are composed of multiple nodes that work together to obtain a response. Each node is responsible for a specific activation function and for assigning weightings [28–32]. Image analysis techniques using artificial intelligence have advanced significantly, opening new possibilities in food quality analysis. Until now, there is no evidence in the literature of the use of artificial intelligence to recognize and control the quality status of dried sweet potato slices under different temperature conditions in the convective drying process. Therefore, an attempt was made to develop effective neural models with the use of machine learning and deep learning to classify dried sweet potato slices subjected to the convective drying process at various temperatures based on their sensory characteristics to obtain the optimal final product.

## 2. Materials and Methods

### 2.1. Material

Sweet potatoes (*Ipomoea batatas*), obtained from a local agricultural enterprise located in the Greater Poland voivodeship, near the town of Kleszczewo, were used as test material for the purposes of this paper. The raw material came from the USA.

### 2.2. Moisture Content

The initial moisture content of tested sweet potato slices, expressed in [%], was determined using an MA 30 weighing machine made by SATORIUS AG (Göttingen, Germany), which enabled us to determine the water content of tested samples with an accuracy of up to 0.05% in accordance with the manufacturer's instructions, at a temperature of 95 °C, as per the PN-A-79011-3:1998 standard. Each sample was subjected to three test series. The average initial moisture content of the test material was $77.88 \pm -0.6\%$.

### 2.3. Convective Drying Procedure

The raw material was sliced crosswise ($2 \pm 0.1$ mm) using a vegetable slicer. The prepared test material (in the form of slices), with an initial weight of $150 \pm 0.1$ g, was placed in a single layer on the shelves of a convective dryer. Then, fresh research samples (sweet potato slices) were subjected to convective drying at four different temperatures: 60 °C, 70 °C, 80 °C, and 90 °C. The velocity of the flow of drying air was v = $1.0 \, \text{m} \cdot \text{s}^{-1}$, as measured using an anemometer. The moisture content of tested samples was measured throughout the drying process and recorded using drying process monitoring software.

### 2.4. Color Measurement

The color of the dried research material was measured using an NH310 m (envisense.eu (accessed on 1 February 2021), EnviSense, Lublin, Poland), which was pressed into close contact with the surface of the material. The white balance of the instrument was calibrated before testing. The amount of light falling on the sensors behind the detector apertures (silicon photodiode) with different colors was measured. L*, a*, and b* values were recorded in the CIE-Lab system [33,34]. The L*, a*, and b* values indicate brightness, redness (+)/(−) green and yellowing (+)/(−) blue, respectively. During the color measurement tests, 15 repetitions were performed. Based on the measured values of L*, a*, and b*, the browning index was determined for each sample [35]:

$$BI = 100 \cdot \frac{\frac{(\text{a}^* \, \cdot \, 1.75 \, \cdot \, \text{L}^*)\text{a}^*}{5.645 \, \cdot \, \text{L}^* \, + \, \text{a}^* \, - \, 3.012\text{b}^*} - 0.31}{0.17} \tag{1}$$

### 2.5. Texture Measurement

Dried sweet potato slices were cut into 1 cm wide strips and subjected to an instrumental texture analysis using a texturometer. Testing was carried out using a TA-XT2i Texture Analyzer (Stable Micro Systems, Godalming, Surrey, UK) [13,36]. Plots of the dependence of resistance posed by tested material on the travel distance of the cutting head were obtained. Based on the collected data, characteristic parameters describing the material texture were determined, such as skin elasticity, maximum shearing force, and work of shearing.

### 2.6. Acquisition of Image of Sweet Potatoes

Macroscopic digital images were acquired using a test rig under constant exposure conditions at the Department of Dairy and Process Engineering. The correct uniformity of illumination of the test sample inside the chamber was verified using a lux meter in accordance with PN-EN 12464-1:2004. Digital images were taken using a NIKON D5100 camera (Kabushiki-gaisha Nikon, Bangkok, Thailand) with a 16.2 megapixel sensor (CMOS sensor 23.1 × 15.4 mm—DX format) and a NIKKOR 28 mm fixed-telephoto lens. The test samples were placed in a shadowless $0.5 \times 0.5 \times 0.5$ m tent, illuminated by visible light

(VIS) with a cold white color temperature of 6500 Kelvin (K), in a room with no natural lighting [37]. To ensure reproducibility of results, the following image exposure parameters were used: aperture—F9, exposure time —125, and sensitivity—IS0-125. The light source used in the test and at the measurement station consisted of 6 lamps: $3 \times 70$ W (600 lm) and $3 \times 60$ W (806 lm). Digital image acquisition produced 180 original 24-bit images with a resolution of $4928 \times 3264$. Image processing resulted in the segmentation of primary images to a frame that included the entire sweet potato discs and a uniformly colored background. Finally, the $600 \times 400$ secondary images were saved in JPEG format and RGB color space.

Images belonging to each class formed equidistant subsets. The subsets were created by pre-extracting features by removing the background and replacing it with a uniform black. Furthermore, data augmentation was applied by rotating each image by 90 degrees to obtain 4 derived images. In addition, data augmentation also consisted of zooming in on a random section or shift of each epoch of the learning sessions [38]. Training, test, and validation sets accounted for 60%, 20%, and 20% of the total number of instances, respectively. Gray-Level Co-Occurrence Matrix (GLCM) discriminants were calculated according to the usage given in the documentation of the Python and MATLAB language function libraries and modules used [15,32,39,40].

### 2.7. Architecture of Used Artificial Neural Networks

Statistica software version 13.3.0 (TIBCO Software Inc., Palo Alto, CA, USA) was used to perform statistical analyses and produce MLP-type artificial neural networks as part of the study. In order to produce deep artificial neural networks, i.e., CNNs, the Python programming language, the Anaconda and IPython environments, and the Spyder editor terminal were used. The study utilized Python modules required for the deep network learning process, such as Numpy, Pandas, Matplotlib, Scikit-image, OpenCV, Tensorflow, and Keras. The operating system used for the deep learning of the artificial neural networks was Ubuntu 20.04 LTS, whereas the Statistica 13.3 package on a Windows 10 operating system was used for statistical calculations.

A Multi-Layer Perceptron neural network was designed using Statistica 13.3, which contained:

- an input layer, i.e., numerical data in the form of 6 texture coefficients such as contrast, dissimilarity, homogeneity, and correlation (indirectly extracted from digital images using the GLCM matrix).
- a hidden layer, for which a range of 5 to 25 neurons was set.
- a set activation function for the hidden layer: Exponential, Hyperbolic Tangent (Tanh), Logistic and output layer type Hyperbolic Tangent (Tanh), Softmax, and Linear.
- an output layer, i.e., numerical data of classes 1–4 on the different drying temperatures.
- a Broyden–Fletcher–Goldfarb–Shanno (BFGS) learning algorithm was set up [41].

A MobileNet type neural network was designed in Python [42], taking a tensor of shape (600, 400, 1) in the case of images expressed in greyscale or of shape (600, 400, 3) in the case of images expressed in RGB (Red Green Blue) as input [43]. The design of the network corresponded to the architecture proposed by Howard et al. [44], the premise of which is the use of separable convolutions, which in practice means the creation of repeating blocks of layers that successively perform the following operations: convolution relative to depth with a $3 \times 3$ convolution window, data normalization, ReLu activation function [45], $1 \times 1$ convolution, data normalization, and ReLu function. After the convolution operations, the edges of the bitmap were additionally padded with zeros (ZeroPadding2D) to maintain the dimensionality of the layer's tensor. After 12 such blocks were applied, a sampling layer (GlobalAveragePooling2D) and a dropout layer were applied to convert random values into zeros in the input tensor. The resulting tensor of shape (1, 1, 1024) was transformed into a tensor of shape (1, 1, 4) by standard convolution. After processing using the softmax activation function, the layer returned a vector of values predicted by the network from the input data.

## 3. Results

### 3.1. Dynamics of Convective Drying Process Applied to Collected Samples

Figure 1 shows the convective drying curves for the four different temperatures used in the drying of test samples (sweet potato slices), i.e., classes 1–4. The total convective drying time at 60 °C (class 1) was 18.8 h, 18 h at 70 °C, 14.8 h at 80 °C, and 13.6 h at 90 °C (Figure 1). Differences between the results of convective drying in each class (1–4) indicated that the higher the temperature, the shorter the drying time. This also translates into noticeable drying shrinkage already in the initial drying phase. The loss of water content and movement of water-soluble compounds caused the stiffness of the cell walls of the sweet potato slices. The greater the loss of water in the sweet potato, the greater the drying shrinkage. This translates into a reduction in the quality of material, as wettability, absorption capacity, shape, and increased hardness all decrease. As the drying temperature increases between classes 1–4, greater changes in shape and shrinkage can also be observed. Considering their final stage of convective drying, where the final weight of the material was assumed to be $49 \pm 0.5$ g, it was determined that the final moisture content of the material in class 1 was 6%, the final moisture content in class 2 was 8.4%, the final moisture content in class 3 was 10.7%, and the final moisture content in class 4 was 7.9%. Thus, the longer the drying time, the greater the reduction of moisture content in research material, which indirectly influences factors such as changes in their shape and the hardness of the test material (classes 1–4).

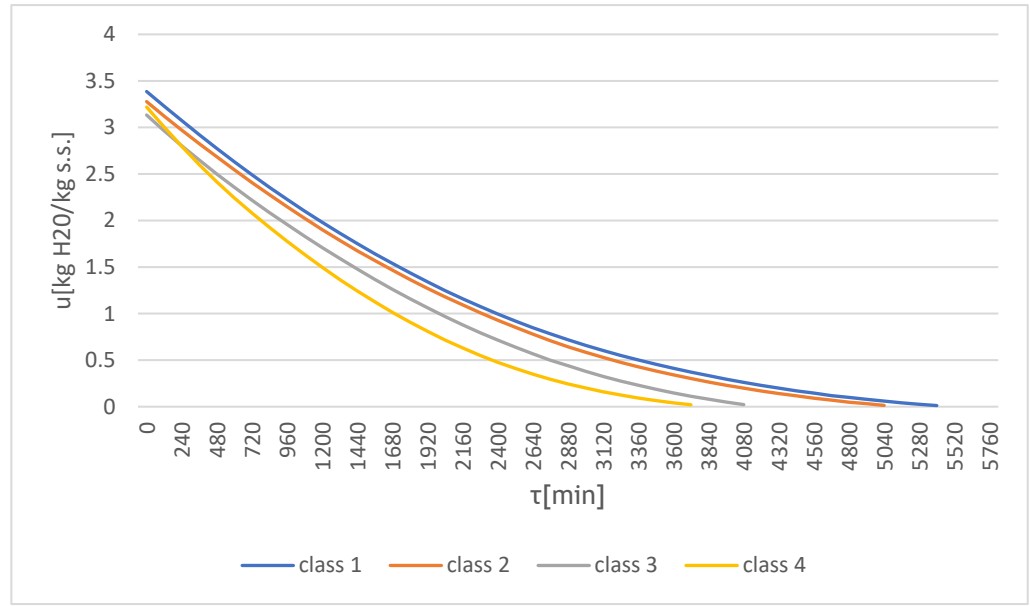

**Figure 1.** Curve of water content per time unit $u(\tau)$ at 60 °C (class 1), 70 °C (class 2), 80 °C (class 3), and 90 °C (class 4).

### 3.2. Statistical Analysis by Mechanical Texture, Color RGB, Color Lab, Texture GLCM

The dried sweet potato slices were subjected to instrumental analysis in order to determine the color and texture discriminants (i.e., shearing force, skin elasticity, and work of shearing) of obtained dried material. Graphs of texture discriminants (Figure 2) illustrate slight differences between classes resulting from each variable, i.e., shearing force, skin elasticity, and work of shearing. The "maximum shearing force" and "skin elasticity" variables proved to be the best at discriminating variables between samples. Analysis of variance showed that class 3 (80 °C) showed statistically significant differences from classes 1 and 2 falling into the second homogeneous group together with class 4.

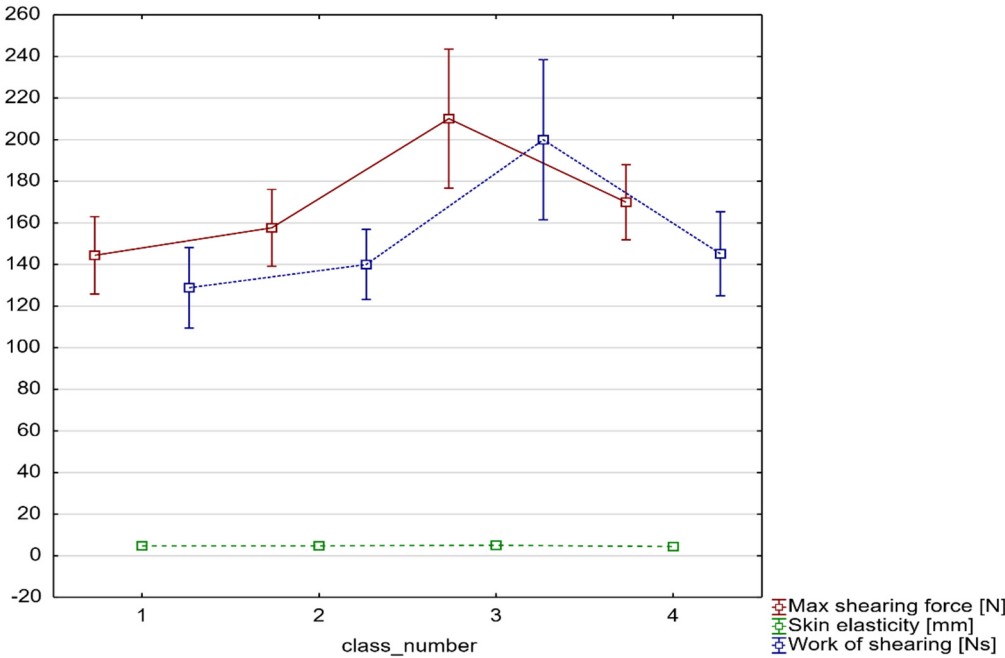

**Figure 2.** Average values of shearing force, skin elasticity, and work of shearing for samples belonging to classes 1–4, which corresponded to drying temperatures of 60, 70, 80, and 90 °C.

In the case of the color parameters expressed in the CIE L* a* b* system (Figure 3), a decreasing trend was observed for the L* parameter describing brightness and an increasing trend was observed for the b* parameter, which represents the yellowing of the tested samples with the increase in drying temperature. The a* parameter characterizing the green–red color axis is not characterized by an unambiguous decreasing or increasing trend. The analysis of variance of the L* variable showed the existence of four homogeneous groups, with the existence of four classes. The neighboring classes in this case overlap to form one homogeneous group for each pair of neighboring classes. The class affiliations for groups a, b, c, and d are as follows: 1[dc], 2[bc], 3[ba], and 4[a]. The decreasing trend indicates a darkening of sweet potatoes as the drying temperature increases. For the parameter a*, two homogeneous groups were observed: 1[ba], 2[a], 3[ba], and 4[b]. This means that classes 2 and 4 are distinguishable at an accepted level of significance. There were no statistically significant differences between classes 1, 2, and 3 and classes 1, 3, and 4. For the parameter b*, the existence of three completely separable homogeneous groups was observed, which, relative to the classes, are as follows: 1[a], 2[b], 3[b], and 4[c]. This means that the change in parameter b* with the increase in drying temperature by each 20 °C resulted in a significant variation in the color of dried slices.

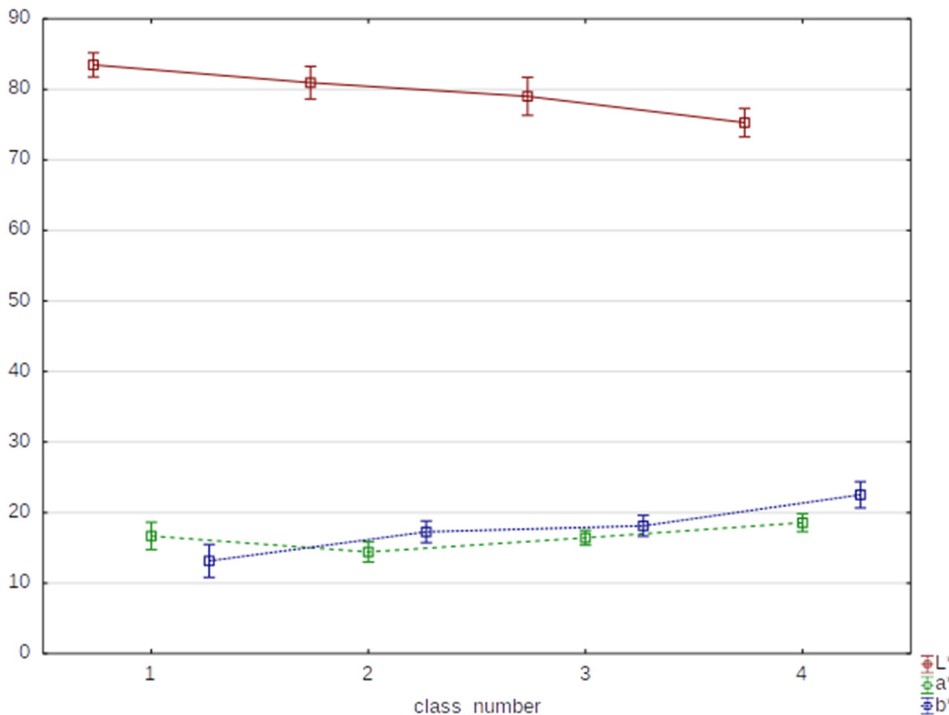

**Figure 3.** Mean values and 0.95 confidence intervals for the color parameters expressed in the CIE L* a* b* system for samples belonging to classes 1–4, which corresponded to drying temperatures of 60, 70, 80, and 90 °C.

The change in color caused by convective drying is explained by two phenomena: Maillard reactions and enzymatic browning. The Maillard reaction is a reaction occurring between amino groups of free amino acids, peptides, or proteins and reducing sugars [46,47]. Products of this reaction are known as Amadori products—compounds with a brown color and a characteristic smell of roasted meat, but which are impossible to describe with a chemical formula due to their diversity caused by the variety of substrates in a given environment [48]. In food processing, decisive factors that positively influence the formation of Maillard reaction products are time, temperature, and water activity. This may indicate a potentially important role of non-enzymatic browning in color changes of convective-dried sweet potato slices.

The second phenomenon affecting the color change of raw plant materials during processing is enzymatic browning. An increase in temperature has a positive effect on the intensity of the browning reaction until the enzyme proteins start to denature. The products affecting the color change are melanin compounds formed in damaged plant cells. One of the enzymes involved in the formation of melanin compounds is polyphenol oxidase. Its action negatively affects the content of phenolic compounds in raw plant materials, which are desirable due to their antioxidant activity [49].

The shape of the graph of the dependence of the browning index value (Figure 4) on the drying temperature indicates that non-enzymatic browning is mainly responsible for the browning of dried sweet potato slices, the intensity of which becomes stronger as the temperature used in the drying process increases. No statistically significant differences in class 1.

An analysis of the variance of texture discriminants calculated from digital images was carried out with the use of different function libraries and programming environments. From among the various variables generated in the MATLAB environment based on the photographs, the six least correlated variables were selected: autocorr, homom, cshad, inf1h, corrm, and cprom. The projection of the aforementioned variables onto the principal component factor plane is shown in Figure 5.

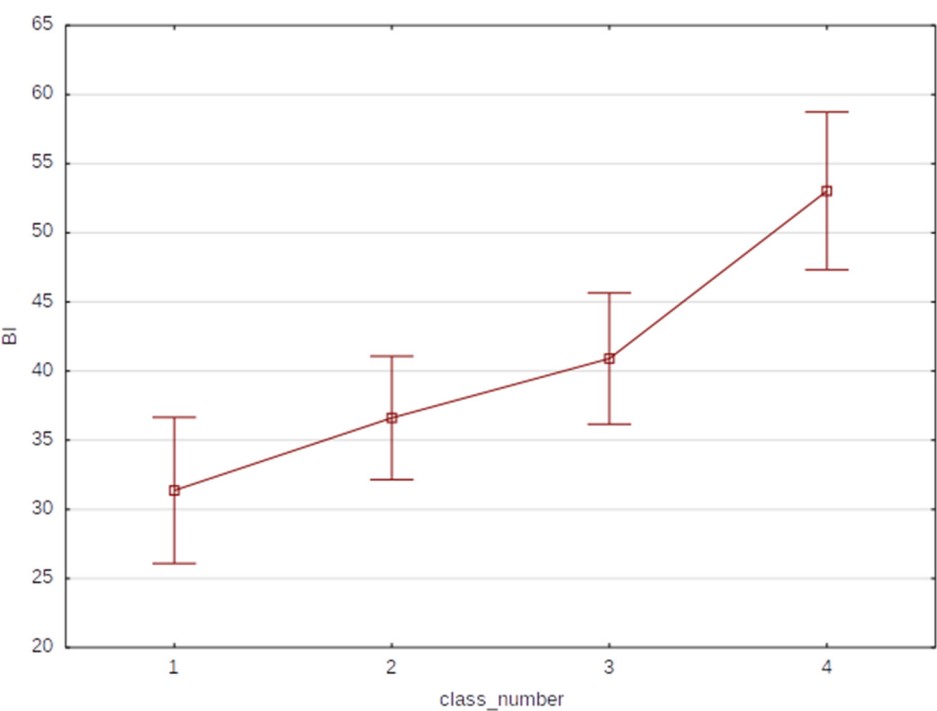

**Figure 4.** Mean values and 0.95 confidence intervals for the BI coefficient for samples belonging to classes 1–4, which corresponded to drying temperatures of 60, 70, 80, and 90 °C.

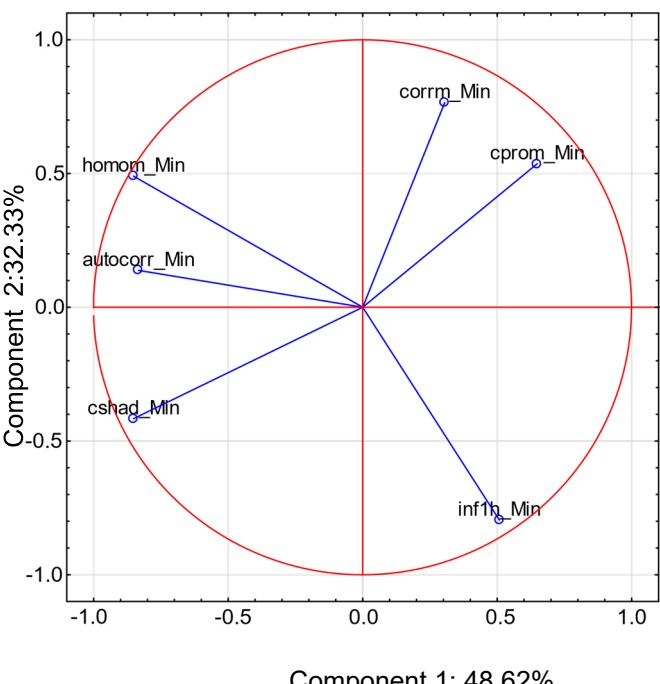

Component 1: 48.62%

**Figure 5.** Projection of selected GLCM texture discriminants calculated in the MATLAB environment onto the principal component factor plane generated using Statistica 13.3.

The analysis of variance of the aforementioned variables showed that autocorr and homom were the differentiating variables. They enabled the separation of three partially separable homogeneous groups for classes 1–4: 1[a], 2[ab], 3[bc], and 4[c]. Image-based texture analysis allowed a more effective distinction than instrumental texture analysis, with class 3 being the significantly different class in both cases. At the same time, class 3 falls into one homogeneous group together with cases from class 4. It can be difficult to effectively

classify a single case due to the overlap between groups, with similar groups b and d not being contiguous groups. For the discriminants calculated using the scikit-image library, the following two homogeneous groups were observed through the analysis of variance: $1^{ab}$, $2^{ab}$, $3^a$, and $4^b$. The results obtained from the texture analysis do not indicate an upward or downward dependence of the variables studied with respect to the drying temperature as is the case with the color analysis.

### 3.3. Classification of Samples Using Artificial Neural Networks

Artificial neural networks were used to classify the samples based on their discriminants. Classification artificial neural network models of densely connected units were generated using the Statistica 13.3 package, using different color and texture discriminants as arguments. These models were characterized by the presence of a single hidden layer. Tables 1 and 2. present the results of learning artificial neural networks for four-class classification on the basis of GLCM discriminants calculated using functions in scikit-image packages or the MATLAB environment. Among the variables returned by the MATLAB environment functions, those least correlated with each other were selected (Figure 5). The top five models were selected based on the quality of testing and validation. The general observable correlation for both types of MLP models considered is low learning quality values, lower than test and validation quality values. This indicates the possibility of random occurrence of correctly classified cases and the uncertainty of the models. This may be related to the low set sizes and the random selection of elements into individual sets. The random selection of sets creates the risk that cases from a significantly different class are aggregated in a single set and that false high learning, testing, or validation quality values occur.

**Table 1.** Results of learning artificial neural networks created in Statistica 13.3, together with parameters of the best models created for the four-class classification on the basis of GLCM features calculated using the scikit-image library (variables contrast, dissimilarity, homogeneity, correlation).

| Name of ANN | MLP 4-3-4 | MLP 4-10-4 | MLP 4-3-4 | MLP 4-6-4 | MLP 4-7-4 |
|---|---|---|---|---|---|
| Quality of learning | 40.476 | 43.651 | 34.127 | 27.778 | 43.651 |
| Quality of testing | 48.148 | 55.556 | 51.852 | 51.852 | 55.556 |
| Quality of validation | 55.556 | 51.852 | 51.852 | 51.852 | 51.852 |
| Algorithm of training | BFGS 21 | BFGS 52 | BFGS 35 | BFGS 9 | BFGS 28 |
| Error function | Entropy | SOS | Entropy | Entropy | SOS |
| Hidden activation | Exponential | Exponential | Tanh | Tanh | Logistic |
| Output activation | Softmax | Linear | Softmax | Softmax | Tanh |

**Table 2.** Results of learning artificial neural networks created in Statistica 13.3, together with the parameters of the best models created for the four-class classification on the basis of the GLCM characteristics calculated using MATLAB (autocorr, homom, cshad, inf1h, corrm, cprom).

| Name of ANN | MLP 6-11-4 | MLP 6-11-4 | MLP 6-10-4 | MLP 6-10-4 | MLP 6-8-4 |
|---|---|---|---|---|---|
| Quality of learning | 66.67 | 32.54 | 34.92 | 37.3 | 36.51 |
| Quality of testing | 55.56 | 40.74 | 40.74 | 44.44 | 40.74 |
| Quality of validation | 44.44 | 48.15 | 51.85 | 48.15 | 48.15 |
| Algorithm of training | BFGS 60 | BFGS 7 | BFGS 4 | BFGS 9 | BFGS 6 |
| Error function | Entropy | SOS | Entropy | SOS | Entropy |
| Hidden activation | Tanh | Logistic | Linear | Logistic | Linear |
| Output activation | Softmax | Logistic | Softmax | Exponential | Softmax |

The learning qualities of presented machine learning models are characterized by incomplete training against learning sets. A different behavior is characterized by deep artificial neural networks, for which 100% of the four-class classification quality was achieved in the case of a network analyzing images expressed on an 8-bit greyscale (Figure 6a).

This leads us to believe that the deep networks used with the MobileNet architecture seek solutions to the classification problem within the boundaries defined by the training set in the considered hyperspace. This increases the reliability of learning, testing, and validation results returned by MobileNet models. The results of learning, testing, and validation are presented as a confusion matrix in Figures 6–8.

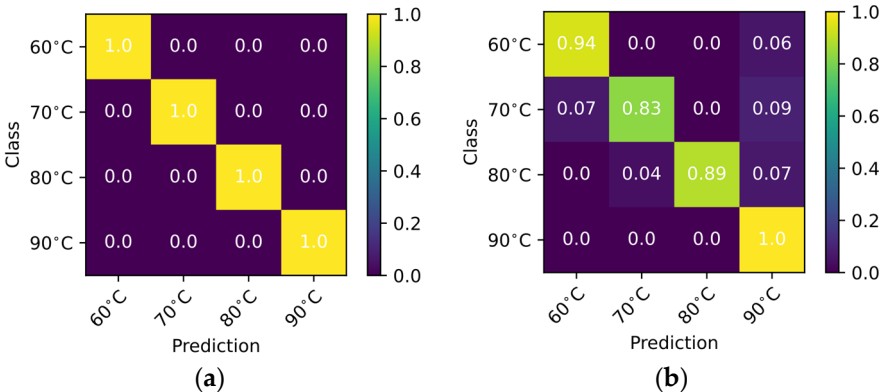

**Figure 6.** Confusion matrix of the MobileNet model for greyscale (**a**) and RGB color space (**b**) calculated from the training set.

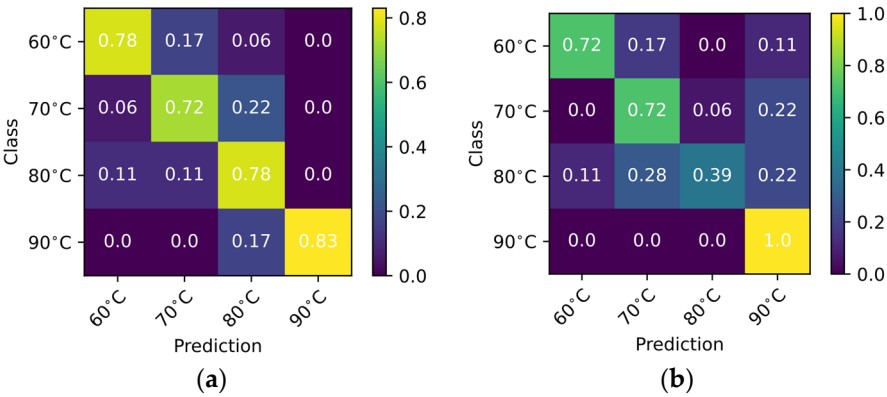

**Figure 7.** Confusion matrix of the MobileNet model for greyscale (**a**) and RGB color space (**b**) calculated from the test set.

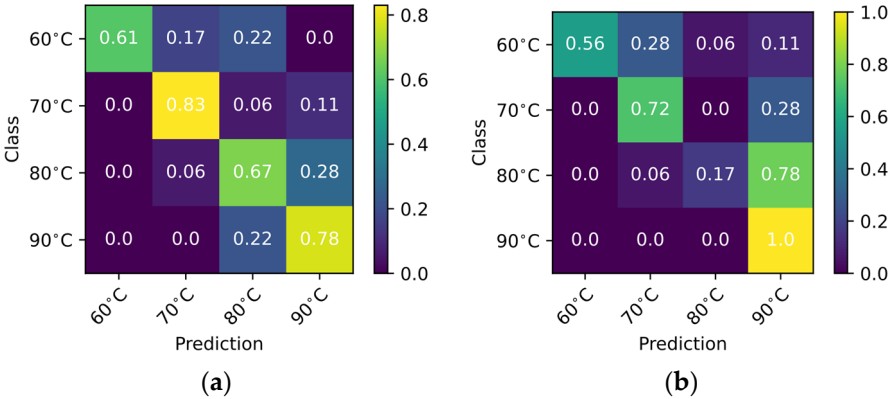

**Figure 8.** Confusion matrix of the MobileNet model for greyscale (**a**) and RGB color space (**b**) calculated from the validation set.

MobileNet (greyscale) for the four-class classification also achieved high test (Figure 7a) and validation quality values (Table 3). In the test set, the best-discriminated class was class 4, of which 83% of the cases were classified correctly (Figure 7a). The remaining

17% of misclassified cases were assessed by the network as class 3. In the test set, 78% of cases belonging to class 3 were classified correctly, while confusions were assessed as belonging to classes 1 and 2 (Figure 7a). Moreover, 72% of cases belonging to class 2 were classified correctly, while 22% were confused with class 3. This indicates that there is some similarity in the texture of cases in classes 2 and 3. Only 6% of cases were falsely assigned to class 1. Whereas, 78% of cases belonging to class 1 were classified correctly and 17% were misclassified as class 2. Only 6% were misclassified as class 3 (Figure 7a). The MobileNet model validation results (greyscale) are more similar to the ANOVA results. The best distinguishable class was found to be class 2, for which 83% of cases were classified correctly, while 6% and 11% were falsely classified as classes 3 and 4, respectively (Figure 8a). Cases belonging to class 1 were correctly classified in only 61%, while 17% and 22% were misclassified as classes 2 and 3, respectively (Figure 8a). The classification results show a high similarity between classes 3 and 4. Cases belonging to class 3 were correctly classified in 67%. Furthermore, 6% were misclassified as class 2 and 28% were misclassified as class 4 (Figure 8a). Cases belonging to class 4 were assigned correctly in 78% and the remaining 22% were misclassified as class 3. The test set in MobileNet (greyscale) indicates the similarity in the texture of sweet potatoes dried at 60–80 °C and the distinctiveness of samples dried at 90 °C. The validation set in MobileNet (greyscale) allows inference of similarity between samples dried at 80 and 90 °C. For comparison, a deep neural network was also created with the MobileNet model, taking into account the color expressed in RGB (Table 3). In the training set, the lowest quality of learning occurred for class 2 and amounted to 83% (Figure 6b). Most often, classification errors resulted from the incorrect assignment of cases in the training set from classes 1–3 to class 4. In the case of the test set, the highest quality of classification occurred for class 4 and amounted to 100%. Only 39% of the class 3 cases were correctly classified and the errors were associated with all other classes. In addition, 72% of cases were classified correctly in class 2, 22% of cases were incorrectly classified in class 4, and 6% as belonging to class 3. Class 1 was also correctly classified in 72%, while 17% and 11% were incorrectly classified in classes: 2 and 4. (Figure 7b). In the validation set, the class 4 cases turned out to be distinct from the others, with 17% of class 3 cases, 72% of class 2 cases, and 56% of class 1 cases correctly classified (Figure 8b).

**Table 3.** Learning results of MobileNettype artificial neural networks against images expressed in RGB or greyscale space.

| Name of ANN | MobileNet-Color | MobileNet-Gray |
|---|---|---|
| Quality of learning | 100.00 | 100.00 |
| Quality of testing | 0.7083 | 0.7778 |
| Quality of validation | 0.6111 | 0.7222 |
| Algorithm of training | Adam | Adam |
| Error function | Entropy | Entropy |
| Hidden activation function | ReLu | ReLu |
| Output activation function | Softmax | Softmax |

The above-described learning results suggest the existence of specific distinguishing characteristics of each of the classes studied. This indicates that the material deforms differently during convective drying depending on the rate of formation of dry skin on the material surface. This is also justified by the increasing trend observed among the results of the instrumental texture analysis. The phenomenon explaining the existence of the observed differences may lie in the thermal decomposition of the hemicelluloses in the plant material during long drying. In this case, the long-dried material would be more susceptible to deformation due to a reduction in the volume of the outer layer compared to material dried at a relatively high temperature, where the hemicelluloses had not decomposed before water drained from the center of the raw material to reach the desired moisture content of the material. Color appeared to introduce noise information

in combination with information on the shape and texture of the dried material. The high variability of the color characteristics can be explained by the simultaneous occurrence of two mechanisms that are opposite in terms of visual effect, namely enzymatic and non-enzymatic browning.

The results presented are comparable or inferior to the performance of artificial neural network classification models reported in literature [32,50–52].

## 4. Conclusions

It was demonstrated that the research can be continued in the qualitative evaluation of sweet potatoes during convective drying. The designed and generated artificial neural networks, especially with the use of deep learning, made it possible to assess the quality, state, and color parameters of dried sweet potatoes based on images (indirectly via the GLCM matrix). Based on tests and analysis of data obtained, we can conclude that there are differences in color and texture between samples of sweet potato convective dried at 60, 70, 80, and 90 °C. The color expressed in the CIE L* a* b* system makes it possible to distinguish sweet potatoes dried at 90 °C from other classes, but involves the possibility of erroneously assigning a sample that does not belong to this class. At the same time, the color expressed in RGB space calculated from digital images does not make it possible to distinguish effectively between the samples tested and, when analyzed using artificial neural networks, makes classification difficult. The positive aspect of using color as a discriminator is that the identification of sweet potato discs dried at 90 °C is subject to the error of falsely assigning the test sample to this class.

A deep artificial neural network of the MobileNet type accepting a greyscale bitmap as input proved to be the most effective tool for dividing dried sweet potato slices by drying temperature. The use of the aforementioned architecture enables a 77% accuracy of the four-class classification. At the same time, binary classification using the MobileNet architecture for greyscale data makes it possible to distinguish with 89% accuracy between sweet potato discs dried at 60 and 70 °C and those dried at 80–90 °C. The use of image analysis and deep artificial neural networks proved to be a more effective method of dividing samples by drying temperature relative to data from a colorimeter or mechanical texture analyzer. This justifies further research into the use of image analysis to evaluate the quality of convective-dried products in online mode. Describing the mechanism of shape and texture change in convective-dried sweet potatoes at different temperatures remains a topic for further research.

**Author Contributions:** Conceptualization, K.P.; methodology, F.A., K.P., J.W. and J.S.; software, F.A. and K.P.; validation, F.A. and K.P.; formal analysis, K.P. and J.W.; investigation, K.P. and J.W.; resources, F.A., J.S. and K.P.; data curation, F.A. and K.P.; writing—original draft preparation, K.P., F.A. and J.W.; writing—review and editing, K.P., J.W., F.A., M.G.-W. and J.S.; visualization, F.A., K.P., D.K. and M.G.-W.; supervision, M.G.-W., D.K. and K.P.; project administration, K.P.; funding acquisition, D.K. and K.P. All authors have read and agreed to the published version of the manuscript.

**Funding:** This research received no external funding.

**Institutional Review Board Statement:** Not applicable.

**Informed Consent Statement:** Not applicable.

**Data Availability Statement:** Not applicable.

**Conflicts of Interest:** The authors declare no conflict of interest.

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
