# Peer review of "Machine and Deep Learning in the Evaluation of Selected Qualitative Characteristics of Sweet Potatoes Obtained under Different Convective Drying Conditions"

_applsci, doi:10.3390/app12157840_

Round 1
Reviewer 1 Report
The manuscript entitled “applsci-1843765” dealing with machine learning has been reviewed. The paper has been nicely written but needs significant improvement. Please follow my comments.
1. Please mention what was the gap in research and add a statement to the introduction.
2. Please briefly introduce the process in the introduction.
3. Add some quantitative results to the abstract.
4. More explanation about Figure 1 “Curve of water content per time” is needed.
5. Add more detail to the conclusion and explain how your findings can support the text.
6. Machine learning has different applications in industries like additive manufacturing. Authors are encouraged to read and add the following new papers related to the application of machine learning in additive manufacturing.
· Relative density prediction of additively manufactured Inconel 718: a study on genetic algorithm optimized neural network models
· Fused filament fabrication of nylon 6/66 copolymer: Parametric study comparing full factorial and Taguchi design of experiments
· Sustainable design guidelines for additive manufacturing applications
Author Response
Dear Reviewer 1,
thank you very much for your valuable comments. I have made the changes as you recommended. This certainly improved and strengthened the substantive aspects of my study.
Response to Reviewer 1 Comments:
Comments and Suggestions for Authors
The manuscript entitled “applsci-1843765” dealing with machine learning has been reviewed. The paper has been nicely written but needs significant improvement. Please follow my comments.
- Please mention what was the gap in research and add a statement to the introduction.
Until now, there is no experience in the literature with the use of artificial intelligence to recognize and control the quality status of dried sweet potato slices under different temperature conditions in the convective drying process. The authors specified the purpose of the work by adding information about determining the optimal final product.
We corrected. Comment taken into account in the text.
- Please briefly introduce the process in the introduction.
The dynamics of the process is conditioned by two mechanisms of water evaporation. The distinction is due to the formation of a crust on the surface of the dried material, which prevents the penetration of water vapor from the dried material into the drying air and heat in the opposite direction. In this context, it is important to distinguish between the evaporation from the surface of the material and the evaporation taking place inside it.
We add reference too: Oke, M.O.; Workneh, T.S. Convective Hot Air Drying of Different Varieties of Blanched Sweet Potato Slices. 2014, doi:10.5281/ZENODO.1097219.
We corrected. Comment taken into account in the text.
- Add some quantitative results to the abstract.
We corrected. Comment taken into account in the text.
- More explanation about Figure 1 “Curve of water content per time” is needed.
We corrected. Comment taken into account in the text.
- Add more detail to the conclusion and explain how your findings can support the text.
We corrected. Comment taken into account in the text.
- Machine learning has different applications in industries like additive manufacturing. Authors are encouraged to read and add the following new papers related to the application of machine learning in additive manufacturing.
- Relative density prediction of additively manufactured Inconel 718: a study on genetic algorithm optimized neural network models
- Fused filament fabrication of nylon 6/66 copolymer: Parametric study comparing full factorial and Taguchi design of experiments
- Sustainable design guidelines for additive manufacturing applications
We add all references. Comment taken into account in the text.
Kind regards,
Krzysztof Przybył

Reviewer 2 Report
1. Author need to mention their exact contribution.
2. existing work limitations did not well analyzed.
3. Color measurement need more detail explanation.
4.how can measure the GLCM matrix?
5. simulation parameters and detail did not well discuss.
Author Response
Dear Reviewer 2,
thank you very much for your valuable comments. I have made the changes as you recommended. This certainly improved and strengthened the substantive aspects of my study.
Response to Reviewer 2 Comments:
Comments and Suggestions for Authors
- Author need to mention their exact contribution.
The authors' contribution at the end of the article is included.
- existing work limitations did not well analyzed.
In line with the opinion of another reviewers, the authors believe that the limitations of the research have been analyzed well.
- Color measurement need more detail explanation.
Authors describe in lines 115-118 about the color measurement and the browning index. To clarify, they add the equation:
We corrected. Comment taken into account in the text.
4.how can measure the GLCM matrix?
The GLCM matrix calculated based on the equations derived by Haralick and by Unser.
We add references:
Unser, M. Texture classification and segmentation using wavelet frames. IEEE Trans. Image Process. 1995, 4, 1549–1560, doi:10.1109/83.469936.
Haralick, R.M. Statistical and structural approaches to texture. Proc. IEEE 1979, 67, 786–804, doi:10.1109/PROC.1979.11328
The authors used the GLCM matrix in their other publications. For the sake of clarification, the authors add more citations to confirm these computational operations:
Przybył K., Koszela K., Adamski F., Samborska K., Walkowiak K., Polarczyk M., Deep and Machine Learning Using SEM, FTIR, and Texture Analysis to Detect Polysaccharide in Raspberry Powders, Sensors, 2021, 21, 5823, DOI:10.3390/s21175823
Przybył, K.; Gawałek, J.; Koszela, K.; Wawrzyniak, J.; Gierz, L. Artificial neural networks and electron microscopy to evaluate the quality of fruit and vegetable spray-dried powders. Case study: Strawberry powder. Comput. Electron. Agric. 2018, 155, 314–323, doi:10.1016/j.compag.2018.10.033.
We corrected. Comment taken into account in the text.
- simulation parameters and detail did not well discuss.
Thank you for your opinion. We corrected. Comment taken into account in the text.
Kind regards,
Krzysztof Przybył

Reviewer 3 Report
This paper discusses the use of deep convolutional artificial neural networks to assess the quality of convective-dried sweet potato slices. This manuscript is well-organized. It can be published after minor revision.
The title should be more specific. It was suggested to change the title to "artificial neural networks in the evaluation of selected qualitative characteristics of sweet potatoes obtained under different convective drying conditions"
There are too many Figures in the manuscript. It is recommended to combine related Figures into one Figure.
Author Response
Dear Reviewer 3,
thank you very much for your valuable comments. I have made the changes as you recommended. This certainly improved and strengthened the substantive aspects of my study.
Response to Reviewer 3 Comments:
Comments and Suggestions for Authors
This paper discusses the use of deep convolutional artificial neural networks to assess the quality of convective-dried sweet potato slices. This manuscript is well-organized. It can be published after minor revision.
Thank you for your opinion.
The title should be more specific. It was suggested to change the title to "artificial neural networks in the evaluation of selected qualitative characteristics of sweet potatoes obtained under different convective drying conditions"
Thank you for your opinion. For the sake of clarity and transparency, the authors have not changed the title.
There are too many Figures in the manuscript. It is recommended to combine related Figures into one Figure.
We corrected. Comment taken into account in the text.
Kind regards,
Krzysztof Przybył

Round 2
Reviewer 1 Report
The paper is ready to publish.
Reviewer 2 Report
All comments are well addresses.